# A New Method for Rapid Subcellular Localization and Gene Function Analysis in Cotton Based on Barley Stripe Mosaic Virus

**DOI:** 10.3390/plants11131765

**Published:** 2022-07-01

**Authors:** Weiwei Chen, Chaolin Huang, Chenmeng Luo, Yongshan Zhang, Bin Zhang, Zhengqing Xie, Mengyuan Hao, Hua Ling, Gangqiang Cao, Baoming Tian, Fang Wei, Gongyao Shi

**Affiliations:** 1Zhengzhou Research Base, State Key Laboratory of Cotton Biology, School of Agricultural Sciences, Zhengzhou University, Zhengzhou 450001, China; weiwei_chen15134@zzu.edu.cn (W.C.); 15638766658@163.com (C.H.); luochenmengzzu@163.com (C.L.); 13938698299@163.com (Y.Z.); xufeizb9456@163.com (B.Z.); zqxie@zzu.edu.cn (Z.X.); haomengyuan1307@126.com (M.H.); caogq@zzu.edu.cn (G.C.); tianbm@zzu.edu.cn (B.T.); 2Henan International Joint Laboratory of Crop Gene Resources and Improvements, Zhengzhou University, Zhengzhou 450001, China; linghua.cqcn@gmail.com; 3National Key Laboratory of Cotton Biology, Institute of Cotton Research, Chinese Academy of Agricultural Science, Anyang 455000, China; 4Department of Biochemistry, National University of Singapore, Singapore 117597, Singapore

**Keywords:** cotton, barley stripe mosaic virus (BSMV), organelle marker, subcellular localization, protein–protein interactions, CRISPR editing

## Abstract

The difficulty of genetic transformation has restricted research on functional genomics in cotton. Thus, a rapid and efficient method for gene overexpression that does not rely on genetic transformation is needed. Virus-based vectors offer a reasonable alternative for protein expression, as viruses can infect the host systemically to achieve expression and replication without transgene integration. Previously, a novel four-component barley stripe mosaic virus (BSMV) was reported to overexpress large fragments of target genes in plants over a long period of time, which greatly simplified the study of gene overexpression. However, whether this system can infect cotton and stably overexpress target genes has not yet been studied. In this study, we verified that this new BSMV system can infect cotton through seed imbibition and systemically overexpress large fragments of genes (up to 2340 bp) in cotton. The target gene that was fused with GFP was expressed at a high level in the roots, stems, and cotyledons of cotton seedlings, and stable fluorescence signals were detected in the cotton roots and leaves even after 4 weeks. Based on the BSMV overexpression system, the subcellular localization marker line of endogenous proteins localized in the nucleus, endoplasmic reticulum, plasma membrane, Golgi body, mitochondria, peroxisomes, tonoplast, and plastids were quickly established. The overexpression of a cotton Bile Acid Sodium Symporter GhBASS5 using the BSMV system indicated that GhBASS5 negatively regulated salt tolerance in cotton by transporting Na^+^ from underground to the shoots. Furthermore, multiple proteins were co-delivered, enabling co-localization and the study of protein–protein interactions through co-transformation. We also confirmed that the BSMV system can be used to conduct DNA-free gene editing in cotton by delivering split-SpCas9/sgRNA. Ultimately, the present work demonstrated that this BSMV system could be used as an efficient overexpression system for future cotton gene function research.

## 1. Introduction

Cotton is the leading natural fiber crop in the world and has been a mainstay of the global economy for over 8000 years [1,2]. In recent years, cotton research has advanced greatly due to the large amount of released cotton genome data [2]. Understanding the molecular and genetic bases of gene functions is of increasing importance to harness their potential to improve the quality, yield, and tolerance abilities of cotton. Gene characterizations at the subcellular level, such as subcellular localization and protein–protein interactions, are a critical element and represent the foundation of gene function studies [3,4]. Conventionally, gene function characterization has relied on bioinformatics predictions, transient transformation, and transgenic techniques. While bioinformatics predictions based on sequence similarity can only provide a guide for gene function analysis, the transient transformation of plant leaves or protoplasts through Agrobacterium infiltration is a relatively common and fast technique to assess gene characterizations in vivo [4,5]. However, because of the heterogeneous and transient expression of certain genes, such a system is not an ideal model for studying genes that are specifically and/or dynamically expressed in cotton. Transgenic plants are optimal materials for studying the precise gene functions in different development periods and growth conditions [5]. Unfortunately, it is difficult and time-consuming to generate stable transgenic cotton plants [4]. Therefore, to accelerate cotton functional genomics research, it is necessary to explore a simple and efficient method that can achieve gene overexpression over a long period time without requiring genetic transformation.

Plant virus-based vectors offer a reasonable alternative to transformation as a powerful tool for stable protein expression in different plant species, and the process is also independent of transformation and tissue culture [6]. The advances that were enabled by plant viral vectors were enabled by the ability of such vectors to derive high levels of gene expression in plants over a short period of time without integration, because of the autonomous replication ability and mobility of these vectors in the infected plants [6]. Many plant viruses, such as tobacco rattle virus (TRV); tobacco mosaic virus (TMV); and potato virus X (PVX), have been successfully applied to express foreign proteins in a wide range of plant taxa [6,7,8]. However, these advances are handicapped by limited cargo capacity, which is typically <1 kb, precluding the use of these vectors for the delivery of full-length gene coding sequences and largely limiting their use to virus-induced gene silencing (VIGS) in plants [9].

Barley stripe mosaic virus (BSMV) is a tripartite positive-sense RNA virus (RNAα, RNAβ, and RNAγ) that is known to infect major crop plants [10]. A three-component BSMV vector, coupling a ligation-independent cloning strategy (LIC) with an *Agrobacterium*-mediated delivery system, was engineered as a VIGS system for high throughput genomic studies in many plants [11,12,13,14]. Recently, it was reported that three-component BSMV vector-mediated overexpression (VOX) can be used as a vector for the expression of small genes such as *iLOV* (330 bp) in wheat roots and leaves [12,15]. However, patchy green fluorescent protein (GFP) expression was observed throughout infect tissues when the GFP (720 bp) was inserted [15].These results indicated that the insertion of larger fragments would limit the systemic spread of BSMV due to viral genome instability [10,12]. To overcome these limitations, Cheuk A and Houde M modified a new BSMV overexpression system comprising a four-component BSMV vector (RNA α, β, γ1, and γ2) that allowed the overexpression of genes of up to 2100 nucleotides and could even achieve the overexpression of a TaALMT1-GFP-*iLOV* fusion gene (up to 2488 bp) [16]. This study also indicated that the four-component BSMV system could be used in a broader range of monocot and dicot plant species to achieve the overexpression of large proteins [16]. However, no study has yet determined whether this system is able to infect cotton and thus achieve stable protein overexpression.

Therefore, we sought to develop a rapid and efficient overexpression system for cotton gene functions research using the four-component BSMV system to overcome the limitations of cotton genetic transformation. Here, we report that this system can infect cotton via seed imbibition and stably express large proteins throughout the plant over a long period of time. We also establish a subcellular localization marker line of eight organelles and present that BSMV can successfully permit the co-localization of two different proteins and facilitate the implementation of bimolecular fluorescence complementation (BiFC) assay to visualize in vivo protein–protein interactions at the subcellular level in cotton. Moreover, our results suggest that the four-component BSMV system can deliver the split SpCas9/sgRNA components to enable systemic gene editing in cotton. Thus, we expect our results to advance the research of cotton functional genomics.

## 2. Results

### 2.1. The Four-Component BSMV System Can Infect Cotton and Stably Express Large Proteins

To investigate the ability of the four-component system to inoculate cotton, we inoculated cotton seeds via seed imbibition with a virus solution containing pCaBS-α, pCaBS-β, pCaBS-γ1, and pCaBS-γ2/pCaBS-γ2:GFP vectors. After 7 days, the green fluorescence of the whole seedlings was detected. As shown in Figure 1A, uneven green fluorescence under UV light could be seen only in the infected plants containing pCaBS-γ2:GFP. Microscopic observations of the cotyledons and stems also revealed the presence of obvious fluorescent signals, compared to those in the control group without GFP (Figure 1B). In addition, we compared the phenotypes and fluorescent signals of the 4–6 true leaf stages between non-infected and infected plants. The results indicated that the BSMV did not affect the normal growth of cotton (Appendix A) and that the fluorescent signal could still be detected even one month later (Appendix A). Taken together, these results indicated that the four-BSMV system could successfully infect the different tissues of cotton and that the targeted gene could be constitutively expressed for at least one month without a negative influence on plant growth.

Additionally, two vectors, pCaBS-γ2:*PM*-*GFP* (1593 bp, plasma membrane location gene) and pCaBS-γ2:*MT*-*GFP* (2340 bp, mitochondria location gene), were constructed to test the maximum capacity of inserted fragments in cotton. The presence of localization signals in cotyledons and roots was observed (Figure 1C). The semi-quantitative PCR and real-time PCR revealed that the GFP was overexpressed in the infected plants but not in the control plants. The expression level in the roots was higher than that in the cotyledons, and the expression level decreased as the inserted target gene fragment increased (Figure 1D,E). This result may be related to the cargo capacity of the viral vector. The results above demonstrated that the BSMV system could express at least 2340 bp in cotton. Meanwhile, based on the gene-length density of *G. hirsutum*, more than 88% of the cotton genes can be overexpressed via this method (Appendix A).

### 2.2. Establishing BSMV-Mediated Organelle Marker Lines in Cotton

The outstanding results of the four-BMSV system encouraged us to establish endogenous-expression subcellular localization marker lines in cotton. First, we isolated eight cotton organelle marker genes and confirmed that they could accurately localize in cotton protoplasts. Homologous marker genes in cotton that were related to existing organelle markers in Arabidopsis or rice were amplified (Table 1). These marker genes localized in the nucleus, endoplasmic reticulum, plasma membrane, Golgi body, mitochondria, peroxisomes, tonoplast, and plastids. Then, the localization of these eight marker genes was confirmed in cotton protoplasts using the transient transformation system. Under a confocal laser-scanning microscope, all the marker genes were detected with fluorescent signals, and different organelles exhibited different fluorescent signals that were consistent with the characteristics of the organelles (Appendix A). However, protoplast preparation and transformation were found to be time-consuming and inefficient, and root protoplast extraction was not easy to achieve, compared to leaf protoplast extraction, due to the former’s lower quantity and quality. In addition, the auto-fluorescent background greatly interfered in the detection of subcellular localization signals. Moreover, due to the characteristics of the transient transformation of Ti vectors, the obtained GFP signal was weak and transitory, especially in the membrane (Appendix A). Therefore, we required a better system to accomplish the goal of establishing steady subcellular localization marker lines.

Subsequently, we established a set of vectors based on the BSMV overexpression system to use for the subcellular localization. We immersed the cotton seeds in a virus solution containing different organelle markers and then observed the fluorescent signals in the root tips and lower epidermis of the cotyledons. As shown in Figure 2A, the plasma membrane, tonoplast, and endoplasmic reticulum exhibited an extended, continuous, and sheet-like morphology (Figure 2A). The fluorescence signals of the plasma membrane and tonoplast were similar in the roots and cotyledons, where a fluorescent signal surrounded the whole cell on the membrane. However, the membrane localization signal of the tonoplast was recessed toward the middle, while the plasma membrane was evenly distributed throughout the cell membrane (Figure 2A, PM and TP). For the endoplasmic reticulum, there was an irregular reticular fluorescent signal scattered in the cells in both the roots and cotyledons (Figure 2A, ER). On the other hand, the plastids, peroxisomes, mitochondria, Golgi body, and nucleus shared a characteristic punctuated fluorescence pattern as a result of their relatively small sizes and multiple copy numbers per cell, except for the nucleus (Figure 2B). The fluorescence signals of the plastids in the chloroplasts were separated from each other in the cotyledons, while there was a continuous fluorescence signal filling almost the whole cell in the roots that was distinctly different from the obvious chloroplast morphology in the cotyledons (Figure 2B, PL). In both roots and cotyledons, the nucleus was observed as a relatively large circle that was generally biased toward the cell side (Figure 2B, NU). However, the peroxisomes were more scattered in the cell (Figure 2B, PR). The mitochondria had different shapes in the roots and cotyledons. Most mitochondria in the roots assumed the form of dots, while those in the cotyledon were flatter and more numerous (Figure 2B, MT). The Golgi body was similar to the mitochondria and featured a large number of spot-like markings in the cotyledons, but it was shorter in the roots with a flat shape (Figure 2B, GB). These results were consistent with the subcellular localization using cotton mesophyll protoplasts, again demonstrating the reliability of the localization of endogenous organelle genes. In addition, the results after infecting the cotton with viral vectors were obtained much more clearly and easily than the results that were obtained in protoplasts. Additionally, the localization results were obtained for different tissues, which was useful for exploring the functions of tissue-specific expressed genes.

### 2.3. BSMV Allows Stable Gene Expression in Cotton

Considering the mobility and replication of the virus, we believe that the BSMV system can detect the expression of targeted genes over a long period of time. Bile acid: sodium co-transporter (BASS) refers to a family of sodium-dependent transporters that are localized on the plastid membrane [24]. To verify that the four-component BSMV system is suitable for the stable expression of functional proteins in cotton, we constructed a pCaBS-γ2:*GhBASS5*-*GFP* viral vector and then infected cotton seeds to achieve the overexpression of *GhBASS5* in cotton. As shown in Figure 3A, the localization signal of GhBASS5 was successfully detected in the true leaves of the cotton 4 weeks after inoculation. A further time series analysis of the experimental group revealed that the auto-fluorescence of the chloroplast gradually weakened and eventually disappeared as time went on, but there was little change in the GFP fluorescence signal (Figure 3A, Appendix A).

To further explore the detailed changes of the fluorescence signal in the time series, we selected four regions to carry out data statistics on the fluorescence intensity of GFP, CHLO and PMT (Bright). The results showed that the white field (PMT) fluorescence intensity was stable at 32, and the initial intensity of chloroplast auto-fluorescence (CHLO) was stable at about 100, which was much lower than the GFP signal. Further, the intensity value would rapidly weaken over time and disappear after 360 s. The initial value of the GFP signal (about 190) clearly displayed the positioning result. With the passage of time, although the overall intensity tended to weaken, it still maintained at 170, highlighting the stability of the GFP fluorescent signal (Figure 3B). Subsequently, we photobleached part of the fluorescence signal of one plastid and then monitored that signal under a normal fluorescence intensity. After 10 minutes, the GFP signal of the quenched area was recovered with no difference to the signals of other non-treated plastids (Figure 3C,D). Thus, we believe that the continuous replication of the virus allowed GhBASS5:GFP to be produced continuously and thus provide the rapid and stable expression of GhBASS5:GFP protein. This phenomenon may explain why the GFP fluorescence intensity remained at a high level.

### 2.4. BSMV Allowed for Rapid Gene Function Analysis in Cotton

Based on the above results showing that the BSMV system can allow the stable expression of large proteins over a long period of time, we attempted to use the system to study gene functions in cotton. GhBASS5 is a member of the sodium-dependent transporter family that was demonstrated, in our previous study, to localize on the plastid membrane and function in transporting sodium in *Arabidopsis* [25]. We further verified the gene function characterization of *GhBASS5* in the salt tolerance of cotton. Under normal conditions, the BSMV-mediated overexpression of GhBASS5:GFP (GhBASS5-VOX) presented a scattered, large, dot-like plastid fluorescence state in cotton roots (Figure 4A, upper). Interestingly, we found that the scattered, large, dot-like fluorescence state of the plastids yielded integrated fluorescence throughout the whole cell after treatment with 100 mM NaCl (Figure 4A, under), indicating a remarkable change in the subcellular structure at the cellular level during a salt stress response in cotton root. To further explore how *GhBASS5* works under salt stress, we obtained three groups of plants that were treated under different conditions with water, 100 mM and 200 mM of NaCl. The phenotypes of the cotton plants are shown in Figure 4B. We found that GhBASS5-VOX increased the salt-sensitivity of cotton. In addition, we measured the Na^+^ and K^+^ levels in the cotton xylem of the control group (GFP) and GhBASS5-VOX group. The results showed that the Na^+^ content in the stem flow of the GhBASS5-VOX group increased with an increase in the NaCl concentration in the hydroponic nutrient solution (Figure 4C). In contrast, the K^+^ content and K^+^/Na^+^ in the xylem sap flow of the control group decreased. However, under both normal conditions and salt stress, the Na^+^ content in the xylem of cotton overexpressing *GhBASS5* was higher than that in the control group. The K^+^/Na^+^ of the GhBASS5-VOX group was lower than that of the control group under both normal conditions and NaCl treatment. The above results agreed with our data in the two *GhBASS5-OE* transgenic *Arabidopsis lines* showing that *GhBASS5* was involved in Na^+^ transport and that a high level of *GhBASS5* negatively regulated plant salt tolerance [25].

### 2.5. BSMV-Mediated Co-Expression of Two Proteins for Co-Localization and Interactions Studies in Cotton

Besides subcellular localization, protein–protein interactions are another important area of study at the subcellular level for the analysis of gene functions. In general, a co-localization analysis provides an important indication of protein–protein interactions. However, this process is often time-consuming due to co-transformation with protoplasts or transgenic plants. An efficient tool that enables the co-expression of multiple proteins for the rapid analysis of protein interactions in cotton remains a major challenge. To determine whether BSMV could allow the co-expression of two proteins simultaneously, we designed a co-expression strategy using a mixed virus pool considering the limited loading capacity of pCaBS-γ1. We mixed two sets of virus solutions containing two different large proteins in pCaBS-γ2 (Plasma membrane marker (PM-GFP, 1.5 kb) and Golgi marker (GB-GFP, 1.3 kb)) to infect cotton seeds via the seed imbibition method. We then verified whether the two mixed four-component BSMV could infect one cell simultaneously and express two different proteins (Figure 5A,B). As shown in Figure 5C, the plasma membrane distribution of PM and the dot distribution of GB can be seen clearly in the same cells (both in the roots and leaves) at the seedling stage. It can be speculated that this set of viral vectors can be successfully used for co-localization and interaction studies, which could facilitate the functional analysis of cotton genes.

BiFC is a common co-localization technique for verifying protein–protein interactions. However, this method has not been commonly used in cotton due to a lack of efficient over-expression methods. Because the BSMV system could simultaneously express two recombinant proteins in a single cell, we sought to examine whether the BSMV system could be applied to a BiFC analysis in cotton. We employed two proteins that interact on plasma membranes, GhROP4 and GhGGB [26], and conducted pCaBS-γ2:*GhROP4-cYFP* and pCaBS-γ2:*GhGGB-nYFP* vectors (Appendix A). After 7 days of infection, we tore the cotton leaf epidermal cells to observe the fluorescence signal. The YFP fluorescence signals were observed in cotton cells only when GhROP4-cYFP and GhGGB-nYFP were co-expressed by two mixed sets of the four-component BSMV system (Figure 5D). The results suggest that this system could be an efficient tool to visualize in vivo protein–protein interactions in cotton cells.

### 2.6. BSMV Allows the Delivery of CRISPR/Cas9 Reagents for DNA-Free Gene Editing in Cotton

More recently, there has been growing interest in using plant viruses as vectors to deliver CRISPR reagents for transgene-free plant gene-editing. However, the limited packaging capacities of such viruses, which is typically <1 kb, impede the delivery of the large Cas9 cassette (3–kb). Thus, a viral vector system that could express large Cas9 proteins and single-guide RNA (sgRNA) and then spread throughout the entirety of the plant’s tissue (especially the meristematic region for genome editing) would be a game changer. Considering that BSMV can express large sequences and co-deliver two large proteins simultaneously, we sought to determine whether the BMSV system could also deliver the large SpCas9 protein (Streptococcus pyogenes Cas9, 1368 amino acids; full length 4104 bp) and its sgRNA (96 nt) into plants to achieve gene editing. However, the full-length of the SpCas9 gene is 4104 bp, which is well beyond the capacity of the BMSV system. Thus, we split the Cas9 protein at the 714th amino acid to obtain Cas9N (714 amino, full-length 2142 bp) and Cas9C (654 amino acids, full-length 1962 bp) [27]. Because the carrying capacity of the pCaBS-γ1 vector is far less than 2000 bp and only compatible with sgRNA, we inserted both Cas9N and Cas9C into the pCaBS-γ2 vector.

We designed the target site of *GhBASS5* using CRISPR-P2.0 (http://crispr.hzau.edu.cn accessed on 13 January 2017) and inserted the sgRNA into the pCaBS-γ1 vector (Figure 6A,B). To prepare the infection solution, we first mixed the *Agrobacterium* resuspension containing pCaBS-α, pCaBS-β, and pCaBS-γ1:*GhBASS5-sgRNA* in equal proportions, and divided each suspension into two after mixing. Then the pCaBS-γ2:Cas9N and pCaBS-γ2:Cas9C resuspension were mixed with the two solutions, and the two sets of the four-component system were used to infect tobacco leaves. After 7 days, we collected the infected leaves to produce a virus homogenate and mixed the solution to inoculate cotton. As shown in Figure 6C, we infected the cotton cotyledons and then extracted DNA and RNA after 14 days. Based on the 109 bp indel between *GhBASS5-A* and *GhBASS5-D*, the sequences of the *GhBASS5-A* and *GhBASS5-D* in the study were found to be 629 and 520 bp. The two sequences were then enzyme-digested to detect the editing efficiency via a CAPS analysis. The CAPS and RT-PCR results revealed that two homologous *GhBASS5* genes were edited simultaneously in the inoculated and upper system leaves and that all the BSMV four-component and CRISPR reagents were detected, although the efficiencies were significantly reduced in the upper system leaves (Figure 6D,E). The sequencing results showed that the mutation type was dominated by small fragment deletions (Figure 6F). These results suggest that the BSMV system could enable the split-Cas9 protein to function normally and induce gene editing through the co-delivery of split-Cas9 and sgRNA cassettes.

## 3. Discussion

### 3.1. BSMV System Makes It Possible to Carry Out Rapid and Simple Gene Overexpression in Cotton

In recent years, great breakthroughs have been made in the research of cotton transgenic technology, and genetic transformation technology is becoming increasingly mature. However, the low transformation efficiency, time-consuming transformation processes, and strong dependence on receptor genotypes of current methods have greatly limited research progress on the cotton functional genome [4]. The new four-component barley stripe mosaic virus (BSMV) makes it possible to implement gene overexpression in cotton. Most early applications of BSMV in plants have been used for gene silencing in vegetative tissues [28,29,30]. At the same time, the application of the BSMV system in gene overexpression has been explored. Lee et al. [15] found that the BSMV system can stably express a small iLOV protein (less than 500 bp). With the advance of research on the BSMV system, this system’s loading capacity is also improving. Cheuk and Houde [16] reported that a modified four-component BSMV can stably express genes that are greater than 2 kb in many plants. Based on the results, we sought to overexpress genes in cotton by the BSMV system and develop efficient methods to study gene function in cotton. As we expected, we detected uneven green fluorescence in the infected plants (Figure 1A,B, and Appendix A). This indicated that the BSMV had the ability to infect cotton, except for the reported plant host, such as *Secale cereale*, *Triticum aestivum*, *Brachypodium distachyon*, and so on [16].

Compared to methods such as leaf friction and leaf injection, moving the virus through seeds can make gene expression more uniform in different tissues without any virus symptoms [31]. Combining seed imbibition and the virus vector enables non-transgenic gene overexpression experiments in cotton to be carried out smoothly. Unlike the transient overexpression that is caused by an *Agrobacterium*-mediated leaf injection [32], this novel method can also detect gene expression in roots and stems, as shown in Figure 1. Because the virus solution can be preserved over a long period of time, the operation process is much simpler and benefits from greater repeatability. In addition, the carrying capacity of the BSMV vector in this study was up to 2340 bp, meaning that this vector can be applied to the functional study of most reported genes (over 88%) in cotton (Figure 1 and Appendix A). Overall, the establishment of a cotton overexpression system based on the BSMV system was found to provide convenient gene function analysis in cotton.

### 3.2. Advantages of Establishing Subcellular Localization Marker Lines in Cotton

Plant cells have highly complex systems and perform a variety of functions. For example, photosynthesis occurs in chloroplasts, while cell respiration occurs in mitochondria [33]. Subcellular localization is the foundation that is used to reveal gene function. Although bioinformatics prediction and transient expression can be used for subcellular localization analysis, these methods still have many defects [34,35]. The biggest disadvantage of these methods is that they cannot be used to observe protein localization in the living cells of the object itself for most species [36]. The establishment of endogenous organelle protein makers helps to perfect and supplement the traditional methods. In recent years, some species such as *Medicago sativa* and *Oryza sativa* have been used to establish a set of organelle markers based on fluorescent protein [37,38,39]. In this study, we took advantage of the BSMV system and used endogenous proteins to generate a set of organelle markers that were labeled with GFP, which localized in eight different organelles in cotton (Table 1, Figure 2 and Appendix A). The expression of endogenous proteins in the living cells can overcome the limitations that are caused by the expression of exogenous proteins [35].

The use of endogenous cotton proteins can eliminate the risk of cross-species mis-localization [38]. In this study, visualization of each organelle marker suggested that these markers can be used as a comparative standard for determining organelle distribution [39]. Furthermore, considering that the chloroplast auto-fluorescence was red, we can assume that the CHLO channel represented a fusion of another functional gene with the fluorescent protein RFP. When we performed stress treatment, the change in the fluorescence intensity of each channel clarified whether the two functional genes interacted with each other. Meanwhile, the four-component BSMV system allowed the expression of the tested proteins over a long period of time (Figure 3 and Appendix A). In this way, the spatial changes of protein localization could be tracked over a long period of time through cotton organelle marker lines. This is another feature of the BSMV system that is superior to transient expression.

Co-localization is a very simple and fast way to study the function of an unknown gene and the results are more reliable than those of other methods [40]. In this research, the two different morphological organelle markers, plasma membrane marker and Golgi marker, co-localized in the same cell (Figure 5C). The result presented that the BSMV system had the potential for co-localization and also verified the effectiveness of the organelle markers. Furthermore, although it was previously reported that small fragments could be inserted into pCaBS-γ1, we only considered inserting the target gene into pCaBS-γ2 [16]. In this way, we can co-localize two proteins without discussing the size of the targeted proteins. Further, the positive results of co-localization indicated that the BMSV system could also play more roles in gene function research [41]. Moreover, it is also exciting news that BSMV allows the expression of two different proteins in plants, which is of great benefit to the study of gene functions and protein–protein interactions.

### 3.3. The BSMV System Can Be Successfully Used for Gene Function Studies

The establishment of new methods facilitates better scientific research. This new BSMV system has several advantages compared to other transformation systems, including ease of use, reduction in cost, high transformation efficiency, and the ability to express larger genes in different tissues [16]. The function of the overexpressed protein was verified using this system. The time-series analysis of the localization of the *GhBASS5* gene and the xylem stem flow experiment of cotton showed that *GhBASS5* was localized on the plastid membrane and involved in Na^+^ transport (Figure 4), which agreed with our data in the *GhBASS5-OE* transgenic *Arabidopsis* lines [25]. These results indicate that the ease of use of this system allows rapid and efficient transformation to better understand or improve protein function.

It is difficult to validate the protein–protein interactions in vivo due to the challenging transgenic technology that is available for cotton [36]. In our study, we used two methods to explore the possibility of using BSMV to achieve protein–protein interactions in vivo. Using both co-delivery of the two spilt-Cas9 proteins and BiFC technology, the results of gene-editing and BiFC indicated that protein–protein interactions in cotton were successfully performed (Figure 5D and Figure 6). For the BiFC, although we detected the yellow fluorescence signals in the cotyledons, we suspect that the signals would also be detected at the 4–6 true leaves stage because of the mobility of the virus [42]. Notably, we mixed the virus homogenate in equal proportions instead of directly co-injecting *Agrobacterium* containing CRISPR editing elements into *N. benthamiana* (Figure 6C). This method has two main advantages. Firstly, the virus has an exclusive mechanism [43]. When the resuspensions containing Cas9N and Cas9C were mixed and then injected into *N. benthamiana*, the plant recognized the weaker strains and triggered an immune mechanism to resist infection and replication of the other virus component. Secondly, when the two kinds of *Agrobacterium* resuspensions were mixed, Cas9N and Cas9C likely induced fusion in the presence of sgRNA [27]. This phenomenon may have activated Cas9 in the virus homogenate. However, it is difficult to determine whether the targeted mutation was caused by co-delivering split-Cas9 proteins.

Besides the analysis of protein–protein interactions, our results showed that the co-delivery of split-Cas9 and *GhBASS5-sgRNA* cassettes could induce DNA-free gene-editing (Figure 6). This indicated that the BMSV system could also be used as a delivery tool to express CRISPR editing components for DNA-free gene editing in cotton, although the systemic editing efficiency was low (Figure 6D). The delivery of genome-editing reagents by plant RNA virus vectors represents a highly efficient and attractive approach for transgene-free genome editing. Notably, some plant viruses move to meristematic cells and express gene editing components in the cell, which results in the production of mutant seeds and hence overcomes the bottleneck that is caused by genetic transformation and tissue culture [44]. However, the virus vector-mediated expression of Cas9 is challenging due to the vector’s small cargo capacity relative to the large size of Cas9 [9]. BSMV is known to infect the reproductive organs of major crop plants and transmit through seeds [10]. In combination with its high cargo capacity, the new BSMV system would be an ideal vehicle to carry CRISPR components for DNA-free plant genome editing. In this study, we verified that the new BSMV system could systematically overexpress large fragments of target genes up to 2340 bp in cotton (Figure 1C), which is still not enough to carry the commonly used SpCas9. Fortunately, using the methods that were applied in previous studies [27,45], SpCas9 protein could be split into two compact parts at its 714 amino acid position. By combining the split-cas9 strategy and co-expression strategy using a mixed virus pool, catalytic Cas9/sgRNA complexes were assembled in virus infected cells and were able to induce targeted mutagenesis of the homologous genes in allotetraploid cotton (Figure 6). Unfortunately, the systemic editing efficiency was low in the present study (Figure 6D), possibly because it was likely that one set of γ2 component would prevail over the other during virus replication, or because the co-existence of two sets of γ2 vectors instigated a partial or complete loss of the insert via recombination. Therefore, in the future study, we will attempt to deliver small Cas9 proteins, such as SaCas9 (1053 amino acids) using the BSMV system [46]. This simple and efficient BSMV-mediated co-delivery of Cas9 and sgRNA could facilitate transgene-free gene editing in plants.

Although the application of the BSMV system was reported in previous gene function studies [15], the present work represents the first systematic study of BSMV’s application at the subcellular level in cotton, especially the establishment of eight organelle makers and protein–protein interactions in vivo. Our results suggest that the four-component BSMV system combined with our previous Si-VIGS method [47] represents a pair of attractive tools for gene expression and gene function analyses in cotton. This system alleviated the obstacles of using transgenic technology to study cotton functional genomics. However, there are still some limitations to our study. We demonstrated that the BSMV system could express proteins from seeds to seedlings, but we did not research whether the expression could be maintained for a much longer time, possibly into the next generation. We will explore this avenue of research in a future study. On the other hand, we found that the BSMV system could still function at the 4–6 leaf stage when the cotton had been lignified (Figure 1, Figure 4 and Figure 6). Therefore, we expect that the BSMV system would also serve as a versatile and powerful tool for the rapid analysis of gene functions in woody plants, which also do not yet have efficient transgenic technology.

## 4. Materials and Methods

### 4.1. Plant Material and Growth Conditions

The upland cotton (TM-1, the genetic standard line of *Gossypium hirsutum* L.) and *Nicotiana benthamiana* were cultured at 25 °C under a 16 h light/8 h dark photoperiod. The cotton roots and leaves that were infected by the viral system were frozen in liquid nitrogen and stored at −80 °C for RNA extraction. For the salt stresses assay, seedlings at the six-leaf stage were transferred to hydroponics (containing 0, 100, and 200 mM NaCl) for two days. Each treatment was replicated three times.

### 4.2. Plasmid Construction

The organelle marker genes were isolated from cotton seedlings and the coding regions were placed into a binary vector pCAMBIA1300-GFP that was derived from the *Cauliflower mosaic virus* (CaMV) 35S promoter using the restriction enzyme sites *BamHI* and *XbaI*. The three-component BSMV system was generously provided by Professor Dawei Li, China Agricultural University. Next, the four-component BSMV system was transformed, including pCaBS-α, pCaBS-β, pCaBS-γ1, and pCaBS-γ2 vectors, according to Cheuk and Houde [16]. To facilitate efficient cloning, the LIC site was generated, containing an *ApaI* restriction enzyme site into the 3′ terminus of the γb gene. Then, the generated vectors were digested with *ApaI* and removed the *ccdB* gene at the same time. Genes of interest were cloned into pCaBS-γ2 via LIC cloning. Firstly, the fragment of inserts and pCaBS-γ2 vectors were treated with the T4 DNA polymerase, respectively. Then, 100 ng of each treated insert were mixed with 100 ng of the treated pCaBS-γ2 vectors, incubating at 66 °C for 2 min, and cooling at room temperature. The products were then transformed into *Escherichia coli* DH5α. All genes and primers are shown in Appendix A. Schematic representations of the expression construct and virus vector are shown in Appendix A.

### 4.3. Isolation and Transformation of Cotton Protoplasts

The protoplasts were isolated from fresh cotyledons, as described by Li et al. [48] with modifications. Briefly, the cotyledons were cut into 0.5-1 mm pieces and immediately digested in an enzymatic solution: (1.5% (*w*/*v*) cellulase R10; 0.75% (*w*/*v*) macerozyme R10; 0.5 M mannitol; 10 mM MES at pH 5.7; 10 mM CaCl_2_; 0.1% BSA; and 0.05 mM β-Mercaptoethanol) under vacuum treatment for half an hour and at 40 rpm for 3 h. The protoplasts were released via rinsing with an equal volume of W5 solution (154 mM NaCl, 125 mM CaCl_2_, 5 mM KCl, and 2 mM MES at pH 5.7) and filtered through a 40 µm nylon mesh. The pellets were then resuspended in MMG solution (0.4 M mannitol, 15 mM MgCl_2_ and 4 mM MES at pH 5.7) to a density of 2 × 10^5^ cells mL^−1^ to facilitate the plasmid transformation [48]. The PEG-mediated transient protoplasts transformation was performed as described by Yoo et al. [49]. In brief, 15 µg of plasmid DNA was added into 100 µL of protoplasts (2 × 10^5^ cells) and then mixed with 100 µL of PEG solution (40% PEG4000 (W/V), 0.2 M mannitol, and 0.1 M CaCl_2_) by gentle tapping. To complete the transfection, the mixed solution was incubated at room temperature for 30 min in the dark. After incubation, 440 µL of W5 solution was added and mixed well by inverting the tube gently. Next, the protoplasts were pelleted by centrifugation at 500 rpm for 4 min and resuspended in 1 mL of WI solution (20 mM KCl, 4 mM MES, and 0.5 M mannitol at pH = 5.7). Finally, the protoplasts were transferred to multi-well plates and cultured in the dark at 22 °C for 12–18 h.

### 4.4. Agroinfiltration of N. benthamiana and Viral Inoculation

Viral particles that were used for seed imbibition were produced in *N. benthamiana* via agroinfiltration. The pCaBS-α, pCaBS-β, pCaBS-γ1, and pCaBS-γ2 vectors were transformed separately into *Agrobacterium tumefaciens* EHA105, as mentioned previously [14]. Equal amounts of the four *Agrobacterium strains* (OD600 = 0.80) were mixed and incubated for 3–5 h at 28 °C. After infiltration, the *N. benthamiana* samples were cultured in a chamber. At 7 days post-infiltration (dpi), 0.5 g of infected leaves were harvested and ground in 1 mL of 20 mM phosphate buffer (pH 7.2). The homogenate was stored at −20 °C for later use or directly used for viral inoculation. The germinated cotton seeds were used for seed imbibition. Before viral inoculation, the homogenates were diluted to a ratio of 1:20 with distilled water. Then, the seeds were covered by the solution to half of their height for 3 days in a petri dish. Next, the cotton seeds were transferred in vermiculite to enable short-term growth over about 4 days and then transferred to a normal growth environment.

### 4.5. Quantitative RT-PCR Analysis

The tissues were directly frozen in liquid nitrogen for RNA isolation. The total RNA was isolated following the RNeasy Plant Mini Kit (Qiagen) instruction. The RNA samples (2 μg of each sample) were used to generate cDNAs using iScript™ Reverse Transcription Supermix, according to the manufacturer’s instructions. A real-time quantitative RT-PCR was performed on a Light Cycler 480 II (Germany). The 7-day-old seedlings were collected for *GFP* detection with *UBQ7* as a reference gene. The data were then exported as Excel files for analysis. The primers used for the qRT-PCR analysis are shown in Appendix A. The qPCR reaction program was as follows: 95 °C for 5 min, 40 cycles of 15 s at 95 °C, and 58 °C for 30 s.

### 4.6. Fluorescence Imaging

All imaging was obtained using a confocal laser-scanning microscope (LSM880, Axio Observer, ZEISS) with the following settings: Ex 488 nm/Em 525 nm for GFP, and Ex 561 nm/Em 595 nm for chloroplast auto-fluorescence. All images captured the maximum projections of the image stacks. All fluorescence experiments were independently repeated at least 3 times. The time series was about 43 frames (423.2 s) and the long-term exposure of a chloroplast was carried out to simulate photobleaching experiments.

### 4.7. Measurement of Na^+^ and K^+^ Content

Xylem sap was collected to detect the contents of Na^+^ and K^+^ in cotton stems, as described by Myo et al. [25] with some modifications. Briefly, the cotyledons were cut off completely from the base to collect the xylem sap flowing from the incision. Next, 5 μL of the xylem sap exudation was diluted with 5% HNO_3_ solution to 50 mL. The ions content in the diluted solution was determined using an atomic absorption spectrophotometer (Triad Scientific, Inc., Manasquan, NJ, USA).

### 4.8. Detection of Gene Editing

The full-length SpCas9 (4104 bp, encoding 1368 amino acids) was split into Cas9N (714 amino acids) and Cas9C (654 amino acids), following Zetsche et al. [27]. Then, the two split Cas9 proteins were cloned into a pCaBS-γ2 vector via the LIC site. The primers used for subcloning are shown in Appendix A. The DNA and RNA of cotton cotyledons were extracted after 14 days of infection. Based on the alignment of the *gGhBASS5A* and *gGhBASS5D* sequence (Appendix A), we designed primers for a cleaved amplified polymorphic sequences (CAPS) analysis to distinguish between *GhBASS5-A* and *GhBASS5-D* based on a 109 bp indel. The editing efficiency was calculated according to the gray value of the bands that were obtained via the CAPS analysis. A grayscale quantitative analysis was performed using the ImageJ software. Primers used for the detection of gene editing are shown in Appendix A.

## 5. Conclusions

In summary, we developed a simple and efficient method for rapid subcellular localization and gene function analysis in cotton based on the BSMV system. This method provides a subcellular localization marker line of eight organelles in cotton and can co-express two proteins in one cell, in order to carry out protein–protein interactions using BiFC technology. Due to the continuous stability of the BSMV overexpression system, we are able to study the subcellular localization characteristics of the target gene over a long period of time under different stress conditions. Furthermore, this method applies to conduct DNA-free gene editing in cotton by delivering split-SpCas9/sgRNA. Taken together, our results suggest that this novel method is an easy and attractive tool for gene expression and function study in cotton. In the future, we will continue to optimize the BSMV system in cotton and explore whether this system could express proteins for a much longer time, in order to broaden the applications of the BSMV overexpression system, which will help to accelerate the study of cotton gene functions.

## Figures and Tables

**Figure 1 plants-11-01765-f001:**
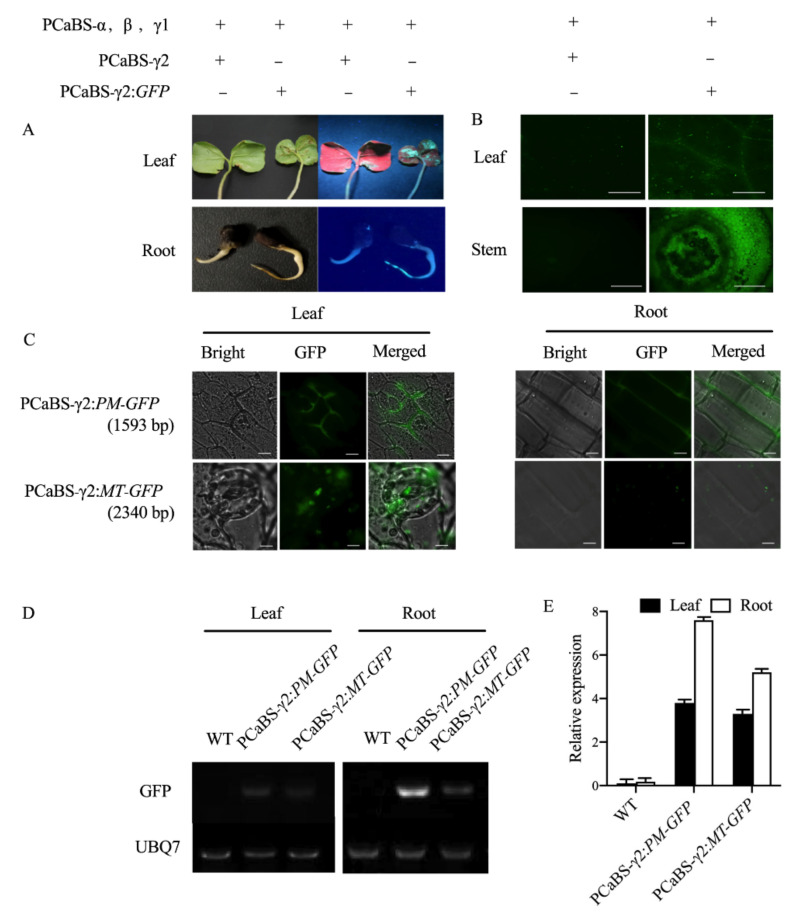
Four-component BSMV system infected cotton via seed immersion and expressed large segments of genes. (**A**). Seed immersion delivery of the BSMV-γ2:*GFP* system infected cotton and expressed GFP; (**B**) microscopic observation of leaves and stems of figure (**A**), respectively. Scale bar is 20 μm; (**C**) microscopic observation of cotton infected with the BSMV-γ2:*PM-GFP* system in row 1, GFP fluorescence showing plasma membrane localization; microscopic observation of cotton infected with the BSMV-γ2:*MT*-*GFP* system in row 2; GFP fluorescence showing mitochondrial localization; leaf-cell infection on the left, and root-cell infection on the right. The scale bar is 5 µm; (**D**) the RT-PCR of GFP in cotton leaves and roots; (**E**) quantification of GFP transcript levels in cotton leaves and roots by qRT-PCR.

**Figure 2 plants-11-01765-f002:**
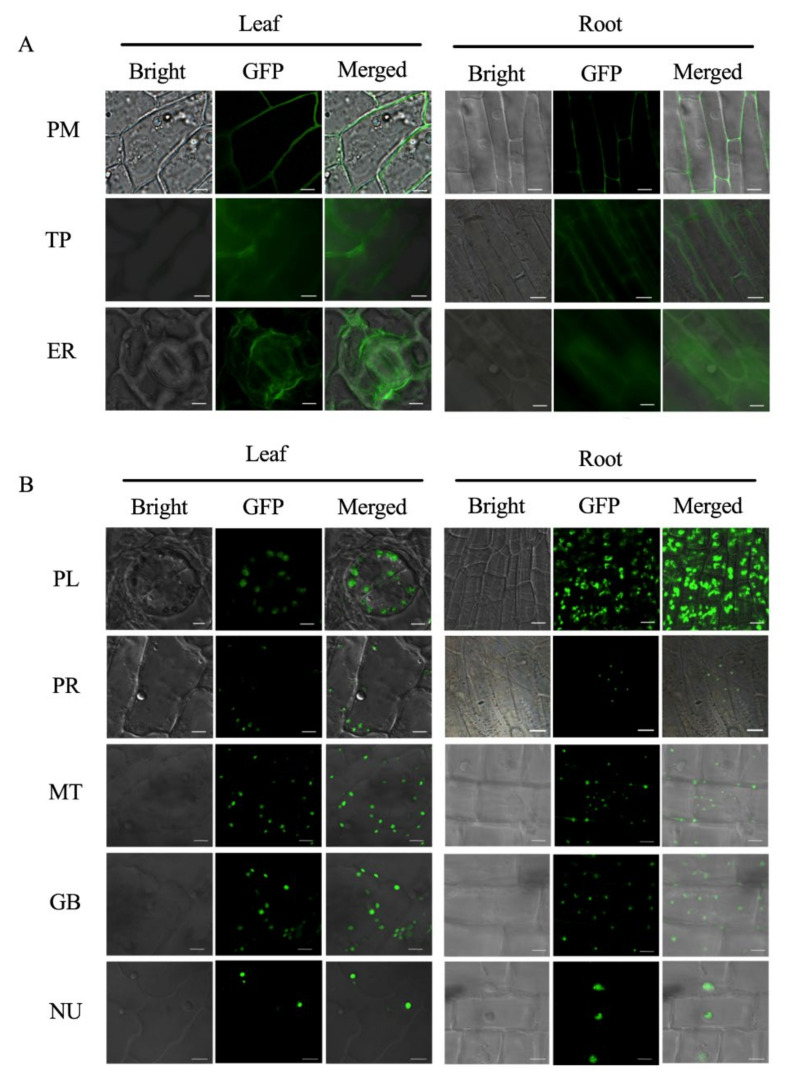
Localization of the eight organelle markers in cotton cotyledons and roots using the four-component BSMV system. (**A**) Subcellular localization of plastids, peroxisomes, mitochondria, Golgi body, and nucleus showing punctate cells; (**B**) subcellular localization of the plasma membrane, tonoplast, and endoplasmic reticulum showing lamellar structures. The scale bar is 5 µm. PM: plasma membrane; TP: tonoplast; ER: endoplasmic reticulum; PL: plastid; PR: peroxisome; MT: mitochondria; GB: Golgi body; NU: nucleus.

**Figure 3 plants-11-01765-f003:**
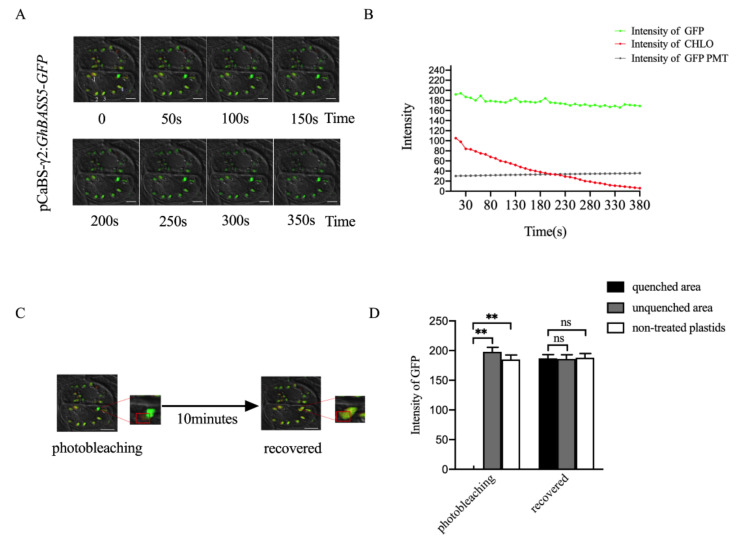
BSMV-mediated stable expression of recombinant protein in cotton. (**A**) Time series of *GhBASS5-GFP* in cotton leaf cells; (**B**) Comparison of the fluorescence intensity of GFP, CHLO, and PMT in the time series; (**C**) Photobleaching experiments, where the red box indicates the quenched area; (**D**) The GFP fluorescence intensity of the treated plastids and non-treated plastids. ** denotes a significant difference at *p* < 0.01 while ns indicates no significant difference, based on a two-tailed Student’s *t*-test.

**Figure 4 plants-11-01765-f004:**
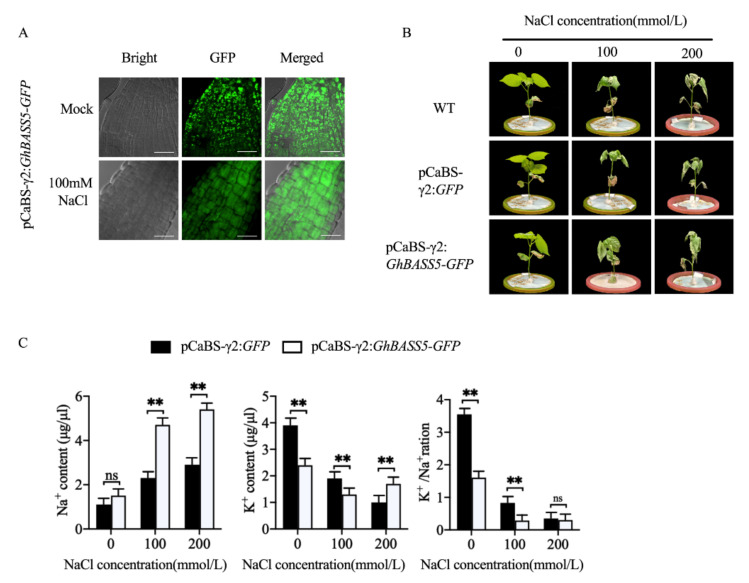
The characterization of *GhBASS5* in the salt tolerance using the BSMV system. (**A**) Fluorescence signal of *GhBASS5-GFP* in cotton root cells under simulated and salt stress; (**B**) phenotypes of the WT, GFP, and *GhBASS5-GFP* plants under simulated and salt stress; (**C**) Na^+^ and K^+^ contents in GFP plants and *GhBASS5-GFP* plants obtained via viral solution immersion under 100 mM salt treatment. The scale bar is 5 µm. ** denotes a significant difference at *p* < 0.01 while ns indicates no significant difference, based on a two-tailed Student’s *t*-test.

**Figure 5 plants-11-01765-f005:**
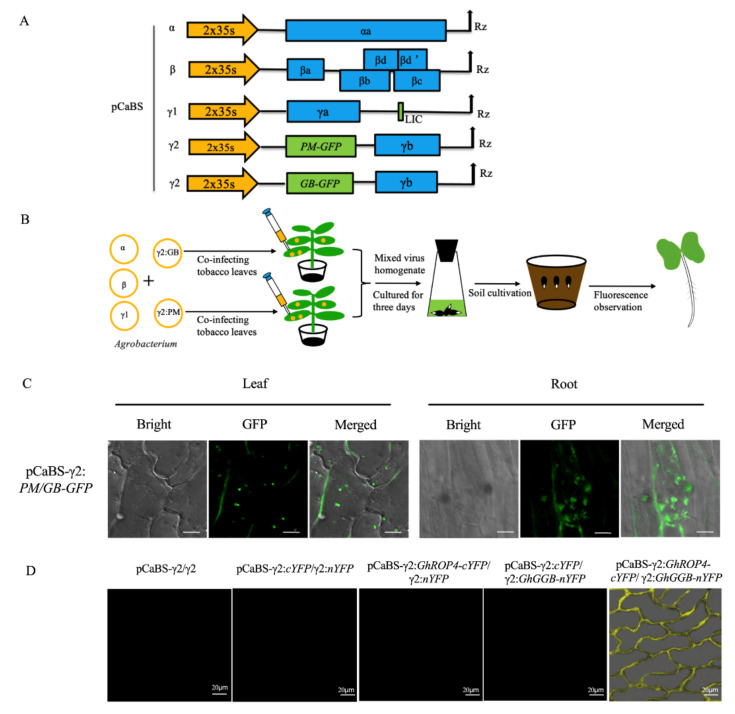
The co-location and protein–protein interactions in cotton mediated by the BSMV system. (**A**) Schematic representation of BSMV recombinant constructs applied to co-express two proteins. The open reading frames (ORFs) of PM-GFP and GB-GFP were in-frame inserted into the gene insertion cassettes of pCaBS-γ2 vectors for simultaneous expression; (**B**) workflow of BSMV medicated co-localization in cotton; (**C**) co-expression of the PM marker and GB marker in one cell. The scale bar is 5 µm; (**D**) in vivo visualization of GhROP4-GhGGB interactions in cotton cells. Each BSMV recombinant construct was mechanically inoculated into cotton as indicated at the top of each image.

**Figure 6 plants-11-01765-f006:**
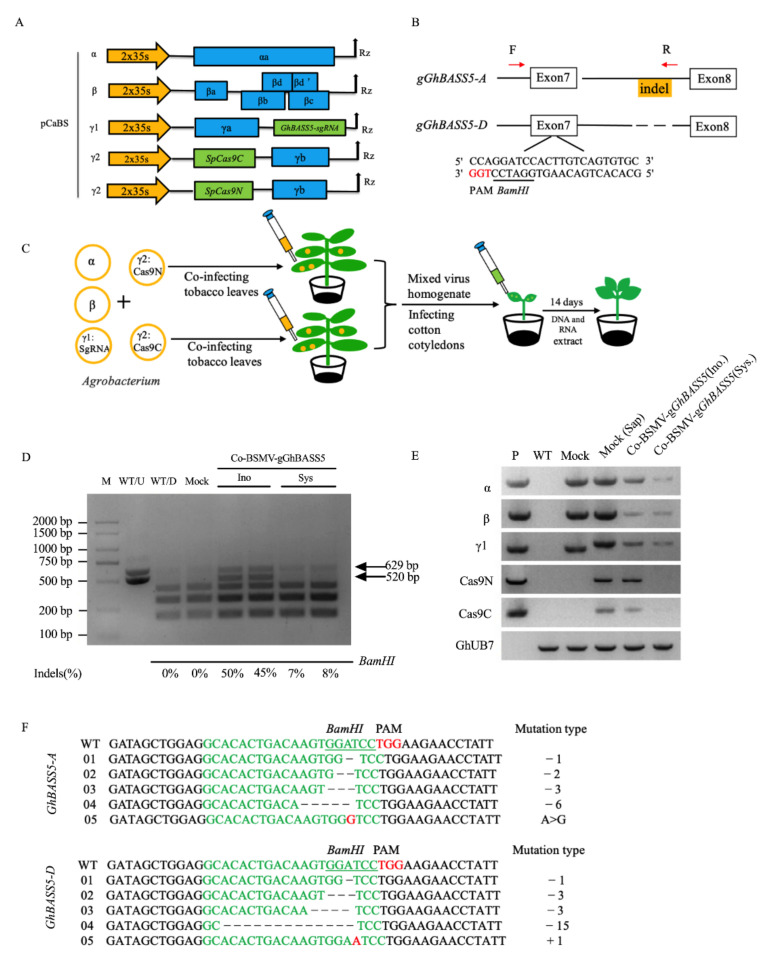
DNA-free gene editing using BSMV-delivered CRISPR/Cas9 system in cotton. (**A**) Schematic representation of the BSMV recombinant constructs applied to the co-delivery of split-Cas9 proteins; (**B**) the gRNA sequences of *GhBASS5*. F and R indicate the positions of primers for CAPS analysis; (**C**) workflow of the BSMV-mediated gene editing system in cotton; (**D**) analysis of the gene editing of *GhBASS5* by CAPS. WT/U and WT/D indicated the undigested bands and digested bands. Mock is the negative control (empty vector). Ino: inoculated leaves; Sys: system leaves (the true leaves); (**E**). expression of each component of recombinant BSMV vectors in inoculated leaves; (**F**). *GhBASS5* gene mutation types analyzed by sequencing.

**Table 1 plants-11-01765-t001:** Selected organelle marker genes and plasmids containing green fluorescent protein.

Organelle	Gene Name	Binary Plasmids	Virus Vectors	Gene Description	Reference
Plasma membrane	*GhPIP2*	pCambia1300-eGFP-PM *	BSMV-γ2: PM-GFP	Transporter activity; aquaporin PIP	Li et al., 2011 [17]
Tonoplast	*GhTIP2*	pCambia1300-eGFP-TP	BSMV-γ2: TP-GFP	Transporter activity; aquaporin TIP	Li et al., 2009 [18]
Endoplasmic reticulum	*GhSPP*	pCambia1300-eGFP-ER	BSMV-γ2: ER-GFP	Aspartic-type endopeptidase activity; integral component of membrane	Tamura et al., 2009 [19]
Plastids	*GhClpD*	pCambia1300-eGFP-PL	BSMV-γ2: PL-GFP	Chaperone protein ClpD, chloroplastic	Dangol et al., 2017 [20]
Peroxisome	*GhAPX3*	pCambia1300-eGFP-PR	BSMV-γ2: PR-GFP	Response to oxidative stress; peroxidase activity	Teixeira et al., 2006 [21]
Mitochondria	*GhALDH2*	pCambia1300-eGFP-MT	BSMV-γ2: MT-GFP	Oxidoreductase activity; metabolic process	Nakazono et al., 2000 [22]
Golgi body	*GhMNS1*	pCambia1300-eGFP-GB	BSMV-γ2: GB-GFP	Mannosyl-oligosaccharide 1,2-alpha-mannosidase activity	Saint-Jore-Dupas et al., 2006 [23]
Nucleus	*GhTAF2*	pCambia1300-eGFP-NU	BSMV-γ2: NU-GFP	DNA-dependent transcription, initiation	Dangol et al., 2017 [20]

* Note PM: plasma membrane; TP: tonoplast; ER: endoplasmic reticulum; PL: plastids; PR: peroxisome; MT: mitochondria; GB: Golgi body; NU: nucleus.

## Data Availability

The data are contained within this article and the Appendix A.

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
