# Peer review of "A New Method for Rapid Subcellular Localization and Gene Function Analysis in Cotton Based on Barley Stripe Mosaic Virus"

_plants, 2022, doi:10.3390/plants11131765_

Round 1

Reviewer 1 Report

 In this manuscript, the authors verified a new BSMV system that could infect cotton through seed imbibition and systemically overexpress large fragments of genes (up to 2340 bp) in cotton. Target gene fused with GFP was expressed at a high level in the roots, stems and cotyledons of cotton seedlings and stable fluorescence signals could be detected in cotton roots and leaves even after 4 weeks. Based on the BSMV overexpression system, the subcellular localization marker lines of endogenous proteins which were accurately located in the nucleus, endoplasmic reticulum, plasma membrane, Golgi body, mitochondria, peroxisomes, tonoplast and plastids were quickly established.

 - Moreover, the authors revealed that overexpression of a cotton Bile Acid Sodium Symporter GhBASS5 using BSMV system discovered that GhBASS5 negatively regulated salt tolerance in cotton by transporting Na+ from underground to shoots. Furthermore, multiple proteins could be co-delivered for co-localization and protein-protein interactions study through co-transformation. In addition, we also confirmed that the BSMV system could conduct DNA-free gene editing in cotton by delivering split-SpCas9/sgRNA. This work presented that the BSMV system could be used as an efficient overexpression system for cotton gene function research.

 - In general, the study is well-performed and revealed a new method for rapid subcellular localization and gene function analysis in cotton based on barley stripe mosaic virusHowever, some revisions are required as shown below;

 - First, the English language of the manuscript should be revised and corrected by native professional or English editing service.

 - The introduction is well written and covered the literature studies and hypothesis. However, the objective of the study should be written in more details at the end of the introduction section.

 - Material and Methods: Details on the methods and assays should be clearly mentioned. Particularly, the PEG-mediated transient protoplasts transformation was performed as described by Yoo et al [18]  (write more details about this transformation protocol? )

 - Results are clear and well-represented. However, the discussion needs more interpretation with the results. It should also be discussed in relation to the present literature.

 -The conclusion should highlight the most significant findings and recommended future studies.

 - The references section needs revisions and should be formatted as per my above-mentioned suggestion

Reviewer 2 Report

Interesting work. Only few changes are required:

Please clarify the objectives of the work sumarizing parragraph lines 92-104.

Clarify cotton cultivar assayed.

Round 2

Reviewer 1 Report

The eview comments have been addressed. 

 The revised version is greatly improved